# Estimation of Chloride Channel Residual Function and Assessment of Targeted Drugs Efficiency in the Presence of a Complex Allele [L467F;F508del] in the *CFTR* Gene

**DOI:** 10.3390/ijms251910424

**Published:** 2024-09-27

**Authors:** Anna Efremova, Yuliya Melyanovskaya, Maria Krasnova, Anna Voronkova, Diana Mokrousova, Elena Zhekaite, Nataliya Bulatenko, Oleg Makhnach, Tatiana Bukharova, Sergei Kutsev, Dmitry Goldshtein, Elena Kondratyeva

**Affiliations:** Research Centre for Medical Genetics, Moskvorechye Str. 1, 115522 Moscow, Russia; melcat@mail.ru (Y.M.); krasnova.m.g.0605@gmail.com (M.K.); voronkova111@yandex.ru (A.V.); diana-mok2000@yandex.ru (D.M.); elena_zhekayte@mail.ru (E.Z.); buben6@yandex.ru (O.M.); bukharova-rmt@yandex.ru (T.B.); dvgoldshtein@gmail.com (D.G.); elenafpk@mail.ru (E.K.)

**Keywords:** cystic fibrosis (CF), complex allele [L467F;F508del], CFTR modulators, therapy

## Abstract

Complex alleles of the *CFTR* gene complicate the diagnosis of cystic fibrosis (CF), the classification of its pathogenic variants, affect the clinical picture of the disease and can affect the efficiency of targeted drugs. The total frequency of complex allele [L467F;F508del] in the Russian population of patients with CF is 0.74%, and in patients with the F508del/F508del genotype, its frequency reaches 8%. This article presents multi-faceted study of the complex allele [L467F;F508del] in a cohort of patients with genotypes [L467F;F508del]/class I (c.3532_3535dup, c.1766+2T>C, W1310X, 712-1G>T), and data for a unique patient with the genotype [L467F;F508del]/[L467F;F508del]. Using the intestinal current measurement method, it was demonstrated the absence of CFTR function for [L467F;F508del]/class I and [L467F;F508del]/[L467F;F508del] genotypes. In intestinal organoids, it was shown that [L467F;F508del] in combination with class I variants and in the homozygotes abolishes the efficacy of both two-component (ivacaftor+lumacaftor; ivacaftor+tezacaftor) and three-component (ivacaftor+tezacaftor+elexacaftor) targeted drugs. When prescribing ivacaftor+tezacaftor+elexacaftor to three patients, they did not have a clinical effect after 6–12 months.

## 1. Introduction

Complex allele carries two or more variants (in the cis position) of the nucleotide sequence [1]. In the last few years, much attention has been paid to complex alleles of *CFTR* gene, which is important for choosing cystic fibrosis (CF) targeted therapy for subjects with such genetic backgrounds and highlighted by decreased sensitivity to CFTR modulators in the presence of additional CFTR cis-variants. For example, the presence of additional cis-variants F87L and I1027T in combination with F508del (complex allele [F87L;F508del;I1027T]) abolishes the sensitivity of F508del to lumacaftor [2]. Complex alleles of the *CFTR* gene also complicate the diagnosis classification of pathogenic variants and affect the clinical picture of CF [3]. There are also known examples when individual variants included in the complex allele are not pathogenic or cause CFTR-related diseases (CFTR-RD) but not CF itself; however, their combination in the cis position leads to CF (if there is also a pathogenic variant present on the second parent allele). In some cases, the “mild” variants enhance each other’s pathogenicity and lead to a “severe” phenotype [4,5]. Rare cases were described when a complex allele reduces the severity of CF (most often, when one of the variants is located in the promoter region of the *CFTR* gene) [6,7]. In Russian patients with CF, the most common complex allele is [L467F;F508del] (c.[1399C>T;1521_1523delCTT], p.[Leu467Phe;Phe508del]). According to the National Register of 2021, it ranks 13th in prevalence, and its total frequency in the Russian population of patients with CF amounts to 0.74% [8]. In addition to the well-studied F508del variant, [L467F;F508del] includes L467F. Baatallah et al. have shown that L467F reduces the level of mature (fully glycosylated) CFTR more than twofold compared to *wtCFTR* [9], while L467F itself is considered a variant with unclear clinical significance (VCV000053246.47—ClinVar—NCBI (nih.gov)).

Previously, we have studied the genotype [L467F;F508del]/F508del in three subjects [10]. Compared to F508del homozygotes, the clinical course of the disease in patients with a complex allele does not differ and corresponds to severe phenotype, and using the intestinal current measurements (ICM) method, we found the reduced function of the chloride channel. In an intestinal organoid model, it was shown that the positive effect of ivacaftor+lumacaftor and ivacaftor+tezacaftor on the restoration of CFTR function was approximately two times lower compared to the F508del/F508del genotype, while due to F508del on the second parental allele, the [L467F;F508del]/F508del genotype retains high sensitivity to the triple targeted drug ivacaftor+tezacaftor+elexacaftor [10]. 

In this study we provide the results of a multi-faceted study of the complex allele [L467F;F508del] in a cohort of patients with genotypes [L467F;F508del]/class I (c.3532_3535dup, c.1766+2T>C, W1310X, 712-1G>T), and the data for a unique patient bearing [L467F;F508del]/[L467F;F508del] genotype. The results include a description of the clinical picture, an assessment of the residual function of the CFTR channel determined by ICM, and data obtained from patient-derived intestinal organoids to assess the sensitivity to CFTR modulators.

## 2. Results

### 2.1. Characteristics of the Clinical Picture of Patients with the Complex Allele [L467F;F508del] and Assessment of Targeted Therapy Efficiency

The study included 5 patients (Table 1), three of them received targeted therapy (Table 2).

Patient 1, male, born in 2016. Genotype: [L467F;F508del]/[L467F;F508del]. CF with pancreatic insufficiency (PI). Chronic purulent obstructive bronchitis. Polypous rhinosinusitis. Protein-energy malnutrition. Chronic infection of the respiratory tract with *S. aureus*. 

The CF was diagnosed at the age of 3 months according to neonatal screening, confirmed by two positive sweat test results (conductivity by the «Nanoduct»—118 and 112 mmol/L), confirmed genetic test—the pathogenic variant F508del in a homozygous state was detected. A child with a low nutritional status receives inhalations with dornase alpha (the second dose per day through nasal cannulas due to polypous rhinosinusitis), pancreatic enzyme replacement therapy, ursodeoxycholic acid (UDCA) preparations and vitamin therapy. The start of therapy with elexacaftor/tezacaftor/ivacaftor + ivacaftor at 6 years of age without changes in clinical and laboratory parameters for 6 months (Table 2), then the drug was cancelled. At the age of 7 years, weight is 17.9 kg, height 116 cm, BMI 13.3 kg/m^2^ (2.9%o). Spirometry indicators: FVC 87%_pred_., FEV_1_ 91% _pred_.

Patient 2, female, born in 2006. Genotype: [L467F;F508del]/c.3532_3535dup. CF with PI. Chronic purulent obstructive bronchitis. Multiple bronchiectasis. Polypous rhinosinusitis. Cirrhosis without portal hypertension (PH). Chronic infection of the respiratory tract with *P. aeruginosa*.

Neonatal screening was not performed (the neonatal screening program in the region started later than the date of birth). From the first months of life, there was visible steatorrhea, delay of growth and development, multiple episodes of bronchitis, and a prolonged cough. According to the totality of clinical symptoms, the diagnosis of CF was suspected, and the child was sent for examination at the age of 2.5 years to the pulmonology department, where, after conducting sweat tests, the diagnosis was confirmed (conductivity by the “Macroduct”—86 and 89 mmol/L). Due to chronic infection of the respiratory tract with *P. aeruginosa* from the moment of diagnosis, he receives inhaled antibacterial therapy and basic therapy (inhalation with dornase alpha and mannitol, UDCA, pancreatin, and vitamins). She has not received targeted therapy. At the age of 16, weight 54.5 kg, height 164 cm, BMI 20.3 kg/m^2^ (44%o). Spirometry indicators: FVC 71%_pred_., FEV_1_ 78% _pred_.

Patient 3, female, born in 2007. Genotype: [L467F;F508del]/c.1766+2T>C. CF with PI. Chronic purulent obstructive bronchitis. Multiple bronchiectasis. Polypous rhinosinusitis. Protein-energy malnutrition. Chronic infection of the respiratory tract with *P. aeruginosa*.

The CF diagnosis was suspected at the age of 4 months according to the clinical picture (steatorrhea, poor weight gain); according to the results of DNA diagnostics, the F508del mutation was detected in a heterozygous state. Neonatal screening and a sweat test were not performed for technical reasons. In 2022, sequencing results revealed two pathogenic variants: L467F in the cis position and c.1766+2T>C in the trans position with the F508del. Due to chronic infection of the respiratory tract with *P. aeruginosa* for 10 years, the patient receives inhalation antibacterial therapy and basic therapy in full (inhalation with dornase alpha and hypertonic sodium chloride solution, pancreatin, vitamins). She has not received targeted therapy. At the age of 16, weight 46 kg, height 166 cm, BMI 16.7 kg/m^2^ (4.5%o). 

Patient 4, female, born in 2013. Genotype: [L467F;F508del]/W1310X. CF with PI. Chronic purulent obstructive bronchitis. Multiple bronchiectasis. Polypous rhinosinusitis. Cirrhosis without PH. Chronic infection of the respiratory tract with *S. aureus*. 

The CF was diagnosed at the age of 2 months according to neonatal screening (data not provided), confirmed twice by a positive sweat test (chlorides of sweat by titration of 80 and 98 mmol/L). The child receives inhalations with dornase alpha, pancreatic enzyme replacement therapy, UDCA preparations and vitamin therapy. At the age of 10 years, weight 27.5 kg, height 132 cm, BMI 15.9 kg/m^2^ (36%o). Spirometry indicators: FVC 116%_pred_., FEV_1_ -110% _pred_. The therapy started with elexacaftor/tezacaftor/ivacaftor + ivacaftor without changes in clinical and laboratory parameters for 6 months (Table 2), and then the drug was cancelled. 

Patient 5, male, born in 2016. Genotype: [L467F;F508del]/712-1G>T. CF with PI. Chronic purulent obstructive bronchitis. Polypous rhinosinusitis. Protein-energy malnutrition. Chronic infection of the respiratory tract with *S. aureus*. 

The diagnosis was determined at the age of 1 month according to neonatal screening (431 ng/mL at a rate of up to 65 ng/mL; retest was not performed), confirmed by a positive sweat test (conductivity by the «Nanoduct»—120 mmol/L) and by DNA diagnostic—pathogenic variants F508del, L467F as part of a complex allele and 712-1G>T in the transposition. A child with a low nutritional status receives inhalations with dornase alpha (the second dose per day through nasal cannulas due to polypous rhinosinusitis), pancreatic enzyme replacement therapy, UDCA preparations and vitamin therapy. The start of therapy with elexacaftor/tezacaftor/ivacaftor + ivacaftor at 6 years of age without changes in clinical and laboratory parameters during the year (Table 2). Due to the progression of polypous rhinosinusitis, surgical treatment was performed after a year of therapy with a targeted drug. The targeted drug has been cancelled. At the age of 7 years—weight 20 kg, height 116 cm, BMI 14.6 kg/m^2^ (24%o). 

When examining the microflora of the respiratory tract in all patients during the follow-up period, no MRSA, Chronic *Burkholderia cepacia*, Nontuberculous Mycobacteria, *Stenotrophomonas maltophilia*, or Gram-negative bacteria were detected. There were no complications such as diabetes, osteoporosis, pneumothorax, hemoptysis, salt loss syndrome, ABLA and oncological diseases among the patients. This data is not included in the Tables.

Elexacaftor/tezacaftor/ivacaftor + ivacaftor treatment was received by patients 1 and 4 for 6 months and patient 5 within 12 months (Table 2). In all patients, there were no positive changes in the sweat test; in patients 1 and 4, an increase in the sample index was registered after 6 months of therapy. The number of exacerbations did not change, there were no positive changes of chronic rhinosinusitis, and patient 5 required surgical therapy for nasal polyposis. No adverse side effects were detected in all three patients during the targeted therapy period.

### 2.2. Evaluation of the Functional Activity of Ion Channels on the Surface of the Intestinal Epithelium by the Intestinal Current Measurements (ICM) Method

ICM has shown that short-circuit current density (ΔISC) in patients with the [L467F;F508del]/class I genotype in response to amiloride (ENaC channel inhibitor) was −5.21 ± 0.98 µA/cm^2^ (Figure 1 and Figure 2 and Table 3). The change in ΔISC in response to forskolin (CFTR channel stimulation) was 1.23 ± 0.48 µA/cm^2^ (Table 3), corresponding to a severe genotype. In response to the introduction of histamine (CaCCs channel), ΔISC shows negative changes (Figure 2, [L467F;F508del]/class I genotypes), reflecting the outflow of potassium ions from the cells. At the same time, the current density amounted to 5.23 ± 0.67 µA/cm^2^. Conclusion: the test indicates the absence of a CFTR function for [L467F;F508del]/class I genotype.

The ΔISC in a patient with genotype [L467F;F508del]/[L467F;F508del] in response to the introduction of amiloride was −5.5 ± 1.54 µA/cm^2^ and practically did not differ from the patients with genotype [L467F;F508del]/class I (Figure 1 and Figure 2 and Table 3). Change in ΔISC in response to the introduction of forskolin was 4.17 ± 0.2 µA/cm^2^; the indicators correspond to the severe genotype but were higher than in the group of patients with the [L467F;F508del]/ class I genotype. In response to histamine administration, ΔISC changes in the negative direction (Figure 2), which reflects the outflow of potassium ions from the cells. At the same time, the current density was 7.0 ± 1.06 µA/cm^2^. Conclusion: the test indicates the absence of a CFTR function for [L467F;F508del]/[L467F;F508del] genotype.

### 2.3. Evaluation of the Effectiveness of CFTR Modulators in Intestinal Organoids Derived from Patients with [L467F;F508del]/Class I and [L467F;F508del]/[L467F;F508del] Genotypes

Using intestinal organoids carrying the genotypes [L467F;F508del]/[L467F;F508del] and [L467F;F508del]/class I, we studied the effectiveness of the potentiator VX-770 (ivacaftor), corrector VX-809 (lumacaftor) and the combined action of VX-770+VX-809 (ivacaftor+lumacaftor), VX-770+VX-661 (ivacaftor+tezacaftor) and VX-770+VX-661+VX-445 (ivacaftor+tezacaftor+elexacaftor). The control groups were presented with organoid cultures with previously well-studied genotypes [L467F;F508del]/F508del [10] and F508del/F508del [12]. Forskolin was used in the concentration range of 0.128–5 μM depending on the “severity” of the *CFTR* genotype. In the case of “severe” *CFTR* variants, organoids were treated with forskolin at a high concentration (5 μM), at which the CFTR channel of the apical membrane is fully activated.

Stimulation with forskolin without CFTR modulators allows us to assess the residual functional activity of the CFTR channel. When evaluating the residual CFTR function, the expected results were obtained for all seven cultures, including controls. In general, the residual CFTR function was either absent or insignificant (Figure 3 and Figure 4). But even with such insignificant responses, a consistent pattern was observed: the absence of the F508del variant in the study group with genotypes [L467F;F508del]/[L467F;F508del] and [L467F;F508del]/class I leads to a complete lack of response to forskolin (Figure 3), no change in the size of organoids occurs and AUC (Area Under the Curve) values do not exceed 71.1 ± 44.3 arbitrary units (a.u.) for [L467F;F508del]/[L467F;F508del] organoids. The presence of F508del on one parental allele (control [L467F;F508del]/F508del) already increases the response to 219.8 ± 78.4 a.u., and in a homozygous F508del culture the responses are 382.3 ± 106.4 a.u.

The use of VX-770, VX-809, as well as double combinations of VX-770+VX-809 (ivacaftor+lumacaftor) and VX-770+VX-661 (ivacaftor+tezacaftor) did not produce even a minimal positive effect on the restoration of CFTR function in the study group (Figure 3 and Figure 4). In the control culture [L467F;F508del]/F508del, swelling of organoids was observed upon treatment with these CFTR modulators, but the responses did not exceed 1000 a.u., unlike F508del/F508del organoids, in which CFTR function was restored, especially by VX-770+VX-809 and VX-770+VX-661combinations.

It is known that the presence of the F508del variant, even on one parental allele, is the basis for prescribing the triple-targeted drug ivacaftor+tezacaftor+elexacaftor (VX-770+VX-661+VX-445) to patients with CF. When exposed to a triple combination drug, we observed effective restoration of CFTR function in control cultures—the organoid response was 2757.1 ± 219.8 and 4718.0 ± 112.0 a.u. for [L467F;F508del]/F508del and F508del/F508del genotypes, respectively (Figure 3), while the size of the organoids increased by more than two times (Figure 4). In the study group, four cultures with genotypes [L467F;F508del]/[L467F;F508del], [L467F;F508del]/c.3532_3535dup, [L467F;F508del]/c.1766+2T>C and [L467F;F508del]/W1310X responded to stimulation with forskolin in the presence of VX-770+VX-661+VX-445, responses were ~500 a.u., the maximum response was observed with the genotype [L467F;F508del]/W1310X (Figure 3), while the size of organoids increases by 15–20% (Figure 4). In the case of the genotype [L467F;F508del]/712-1G>T, the CFTR function was practically not restored.

## 3. Discussion

The L467F variant was described in 2006 in the study of Elahi E et al. [13] and is designated as “Rare”. As part of a complex allele with the F508del variant, L467F was identified in 2016 [14]. However, the studies of molecular mechanisms of L467F and F508del variant’s influence on the level of the CFTR functional protein were performed only later. The study of complex alleles of the *CFTR* gene has become especially relevant since the beginning of therapy of CF patients with CFTR modulators, as one of the severe consequences of the presence of complex alleles is the lack of sensitivity to targeted drugs [1]. In a 2018 study by Baatallah et al. [9], it was found that the L467F variant alone reduces the amount of functional (fully glycosylated) CFTR by two times compared to *wtCFTR* and is not CF pathogenic. However, its presence in the complex allele [L467F;F508del] abolishes the positive effect of the lumacaftor (VX-809) suitable for F508del variant targeted therapy. In 2022, Sondo et al. have shown that the complex allele [L467F;F508del] not only leads to a decrease in CFTR protein synthesis but also to the formation of an exceptionally immature form of CFTR devoid of functional activity, and the treatment with CFTR modulators ivacaftor+tezacaftor+elexacaftor does not increase the amount of mature fully glycosylated form of CFTR, therefore does not affect the activity [L467F;F508del]-CFTR protein [15].

In the Russian Federation, the complex allele [L467F;F508del] is the most common; in patients with the F508del/F508del genotype, its frequency amounts to 8% [10], and, surprisingly, four patients in Russia are homozygotes with this complex allele (not published). For comparison, in the USA, a search for [L467F;F508del] among a cohort of patients with the F508del/F508del genotype found this complex allele in only four patients out of 762, and its frequency was 0.5% (allele frequency ~0.3%) [14]. Previously, we performed a comprehensive assessment of the [L467F;F508del]/F508del genotype in a group of three patients [10]. The main conclusion of our study is that [L467F;F508del] does not affect the clinical manifestations of the disease compared to F508del homozygotes. However, it is accompanied by a loss of sensitivity to the action of two-component targeted drugs, while sensitivity to a three-component targeted drug is preserved (due to the presence of one variant of F508del in the genotype). Similar results were obtained V. Terlizzi with co-authors [16], where the absence of therapeutic responses was observed when ivacaftor+lumacaftor was used in patients with [L467F;F508del] in a compound with the second parental variant of *CFTR* class I or II.

In this work, we studied [L467F;F508del] in a homozygous (one patient) and compound heterozygous position with different class I variants (four patients). The responses to forskolin measured by the ICM method showed the absence of CFTR channel function. Interestingly, the response to forskolin (which was very low in all cases) positively correlated with the amount of F508del in the genotype in the study and control groups: for F508del/F508del was 3.33 µA/cm^2^, for [L467F;F508del]/F508del was lower and amounted to 2.5 µA/cm^2^, and for [L467F;F508del]/class I is only 1.23 µA/cm^2^. Analysis of intestinal organoids showed that presence of [L467F;F508del] in patients with genotypes [L467F;F508del]/class I (c.3532_3535dup, c.1766+2T>C, W1310X, 712-1G>T) and homozygous carrier [L467F;F508del]/[L467F;F508del] negatively affects the effectiveness of both two-component and three-component targeted drugs in vitro treatment. Prescription of ivacaftor+tezacaftor+elexacaftor to three patients exerted no clinical effect after 6–12 months. Previously, the authors also described that the complex allele [L467F;508del] in a heterozygous state with class I variants does not lead to positive clinical changes upon ivacaftor+tezacaftor+elexacaftor treatment (as it was observed during more than 1 year period) [17].

Nevertheless, the widespread complex allele [L467F;F508del] may be considered as potentially amenable to targeted therapy by new CFTR modulators under development since during the FIS assay, AUC values for four cultures ([L467F;F508del]/[L467F;F508del] and [L467F;F508del] in a compound with c.3532_3535dup, c.1766+2T>C, W1310X) were 500–800 a.u. For comparison, when performing FIS assay on intestinal organoids of a Russian patient with a complex allele [S466X;R1070Q] in a compound heterozygous position with a variant of class VII CFTRdele2.3, the AUC values after stimulation with 5 μM forskolin were ~45 a. u., regardless of the presence or absence of a potentiator and correctors, a case which has apparently poor prognosis for being rescued even by enhanced future targeted drugs [18].

## 4. Materials and Methods

### 4.1. Clinical Picture 

The description of the clinical features of five patients was carried out according to the data of the patient register of the Russian Federation for 2021 [8]. The methods of laboratory and instrumental studies were carried out according to the previously described [10].

### 4.2. Intestinal Current Measurements (ICM)

The ICM method was used to confirm cystic fibrosis diagnosis and evaluate the function of ion channels [11]. The study using the ICM method was performed according to European standard operating procedures V2.7_26.10.11 (SOP) [11,19,20,21]. 

The control group (*wtCFTR/wtCFTR*) included 18 healthy volunteers; the comparison group included five CF patients, carriers of the F508del variant in a homozygous state (the absence of complex alleles was confirmed by DNA sequencing) [11].

### 4.3. Forskolin-Induced Swelling (FIS) Assay

The FIS assay was based on protocols developed under J. M. Beekman’s guidance [12,22,23]. A total of 2–4 rectal biopsies were used to obtain a sufficient number of organoids (after three culture passaging). Isolation of crypts from biopsies was preceded by a series of rinses with Advanced DMEM/F12 medium (Thermo Fisher Scientific, Waltham, MA, USA) and PBS (Thermo Fisher Scientific, USA), followed by incubation with 10mM EDTA (Thermo Fisher Scientific, USA). After that, the precipitated crypts were mixed with Matrigel and seeded into culture plates.

The organoids were passaged every 6–7 days The culture medium included the following components: Advanced DMEM/F12 medium, Wnt-3A-conditioned medium (produced in-house [10]), Noggin-conditioned medium (self-produced [10]), R-spondin-1 conditioned medium (produced in-house [10]), mEGF (Prospec), B27 (Life Technologies: Gibco, Waltham, MA, USA), N-acetylcysteine (Sigma-Aldrich, St. Louis, MO, USA), nicotinamide (Sigma-Aldrich, USA), A83-01 (Sigma-Aldrich, St. Louis, MO, USA), SB 202190 (Sigma-Aldrich, St. Louis, MO, USA) and Primocin (InvivoGen, San Diego, CA, USA).

For the FIS assay, organoids were seeded on 96-well plates. After 24 h, the organoids were stained with Calcein AM (BioLegend, San Diego, CA, USA), and then stimulation with 5 µM forskolin was performed. Correctors VX-809, VX-661 and VX-445 (Selleckchem, Houston, TX, USA) were added (3.5 µM) at the stage of organoid seeding, and the potentiator VX-770 (3.5 µM, Selleckchem, USA) simultaneously with forskolin. The intestinal organoids were incubated with forskolin, and the CFTR modulators for 1 h, and the selected fields of view were imaged at 10 min intervals using an Axio Observer 7 Fluorescence microscope (Zeiss, Oberkochen, Germany). Quantitative analysis of the swelling of organoids was carried out using the Image J software (v1.52n state version, NCBI, Bethesda, MD, USA). When plotting the graph (SigmaPlot), the area under the curve of the dependence of the change in the area of organoids on time was calculated using Microsoft Excel (2019). Cultures obtained from patients with the F508del/F508del and [L467F;F508del]/F508del genotypes were used as a comparison group.

## 5. Conclusions

The [L467F;F508del]/class I and [L467F;F508del]/[L467F;F508del] genotypes, the expressed immature glycosylated CFTR protein was not restored by correctors, being insensitive not only to lumacaftor, tezacaftor, but also to a more effective three-component combination with elexacaftor. For the example of three patients, the absence of a therapeutic effect on ivacaftor+tezacaftor+elexacaftor is recorded.

## Figures and Tables

**Figure 1 ijms-25-10424-f001:**
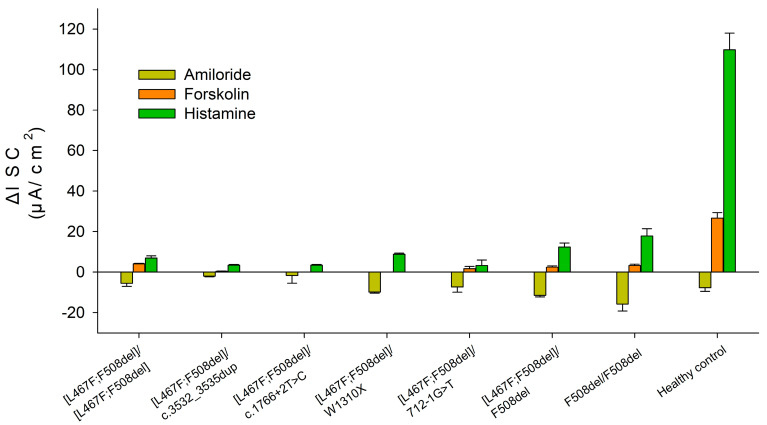
Indicators of short-circuit current density (µA/cm^2^) in response to the administration of stimulants in patients with the complex allele [L467F;F508del] in the genotype. The results for the F508del/[L467F;F508del], F508del/F508del genotypes and healthy control were taken from [10,11].

**Figure 2 ijms-25-10424-f002:**
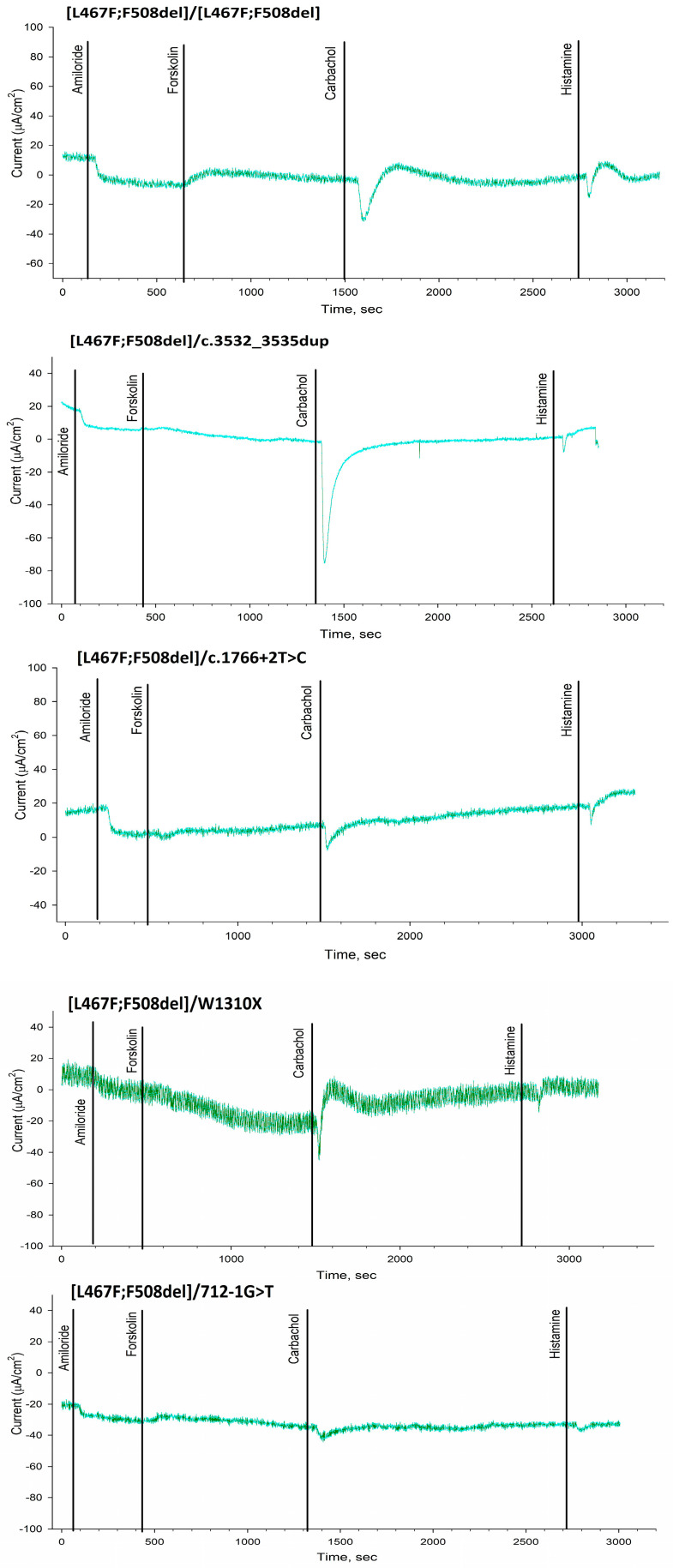
The results of intestinal current measurement in patients with the complex allele [L467F;F508del] in the genotype compared with F508del/[L467F;F508del], F508del/F508del and healthy control.

**Figure 3 ijms-25-10424-f003:**
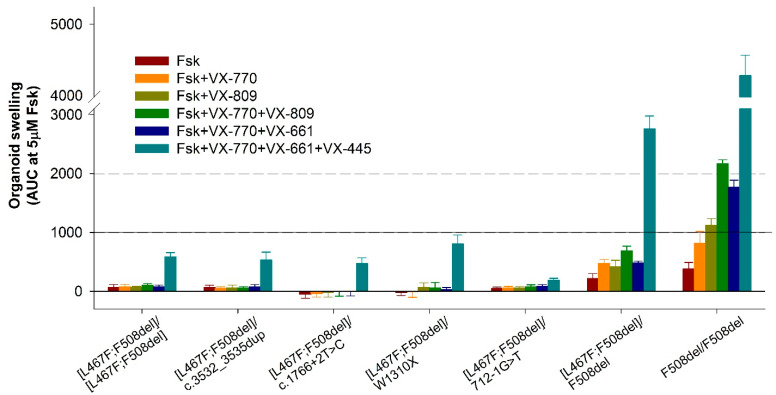
The results of FIS (Forskolin-Induced Swelling) assay in intestinal organoids with genotypes [L467F;F508del]/[L467F;F508del] and [L467F;F508del]/class I compared with control cultures ([L467F;F508del]/F508del and F508del/F508del). A total of 5 μM Fsk (forskolin), VX-770—ivacaftor, VX-809—lumacaftor, VX-661—tezacaftor, VX-445—elexacaftor.

**Figure 4 ijms-25-10424-f004:**
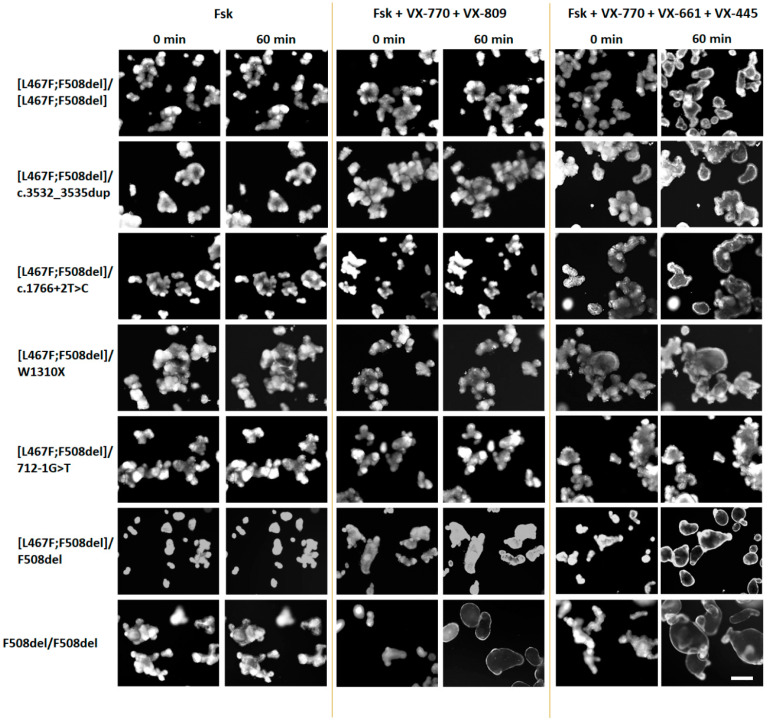
Representative images of intestinal organoids before and after treatment with forskolin (5 μM) and targeted drugs. Scale bar—200 μm.

**Table 1 ijms-25-10424-t001:** Characteristics of the health of the studied patients.

Genotype	[L467F;F508del]/ [L467F;F508del]	[L467F;F508del]/ c.3532_3535dup	[L467F;F508del]/ c.1766+2T>C	[L467F;F508del]/ W1310X	[L467F;F508del]/ 712-1G>T
№	1	2	3	4	5
Age at the time of the study, years	7	16.5	16.5	10	7
Gender m/f	m	f	f	f	m
Age of diagnosis of CF	3 months	2.5 years	4 months	2 months	1 month
Sweat test, mmol/L	118 *	89 *	not performed	98 **	120 *
Fecal elastase-1, μg/g of feces	<15	<15	<15	<15	<15
BMI, kg/m^2^	13.3	20.3	16.7	15.9	14.6
BMI, %o	2.93	44	4.5	36.31	24.46
FEV_1_, %	91	78	not performed	110	not performed
FVC, %	87	71	not performed	116	not performed
Chronic Pseudomonas aeruginosa	-	+	+	-	-
Chronic Staphylococcus aureus	+	-	+	+	+
Liver damage	Without cirrhosis	Cirrhosis without PH	-	Cirrhosis without PH	-
Polyposis of the paranasal sinuses	Sinusitis without polyps	+	+	+	+ Operated on a year after the start of targeted therapy

*—sweat test at diagnosis **—Gibson–Cooke method sweat test.

**Table 2 ijms-25-10424-t002:** Indicators of the effectiveness and safety of targeted therapy.

Indicator	Period of the Targeted Therapy	Patient		
		L467F;F508del]/ [L467F;F508del] Patient 1	[L467F;F508del]/ W1310X Patient 4	[L467F;F508del]/ 712-1G>T Patient 5
BMI (kg/m^2^)	1 day (start)	13.5	-	14.6
	6 months	13.3	15.9	14.6
	12 months	-	-	14.6
Sweat test (mmol/L)	1 day (start)	104 *	89 **	120 *
	6 months	121	95 **	-
	12 months	-	-	114
FVC, %_pred_	1 day (start)	84	120	does not perform the maneuver
	6 months	87.5	116	does not perform the maneuver
	12 months	-	-	does not perform the maneuver
FEV_1,_% _pred_	1 day (start)	85	113	does not perform the maneuver
	6 months	91	110	does not perform the maneuver
	12 months	-	-	does not perform the maneuver
Bronchopulmonary exacerbations	the period of therapy	2–3 exacerbations on targeted therapy, as before the start of therapy	3 exacerbations on targeted therapy, as before the start of therapy	2 exacerbations on targeted therapy, as before the start of therapy
Changes of nasal polyposis and rhinosinusitis	the period of therapy	Without changes	Without changes	operated on due to nasal polyposis
ALT, U/L	1 day (start)	18	24	17
	6 months	25	20	25
	12 months	-	-	18
AST, U/L	1 day (start)	26	30	29
	6 months	31	17	28
	12 months	26	28	30
Total bilirubin, µmol/L	1 day (start)	6.9	7.8	12.3
	6 months	12.7	10.0	11.2
	12 months	-	-	9.8
Systolic blood pressure, mmHg.	1 day (start)	90	95	-
	6 months	95	90	-
	12 months	-	-	-
Diastolic blood pressure, mmHg	1 day (start)	60	65	-
	6 months	60	60	-
	12 months	-	-	-

*—sweat test at diagnosis **—Gibson–Cooke method sweat test.

**Table 3 ijms-25-10424-t003:** Short-circuit current density (ΔISC) as a response to stimulators in a group of patients with cystic fibrosis, M ± m.

Genotype		Amiloride	Forskolin	Histamine
[L467F;F508del]/ [L467F;F508del] Patient 1	Biopsy № 1	−4	4	8.5
	Biopsy № 2	−4.5	4	5.5
	Biopsy № 3	−8	4.5	7
	M ± m	−5.5 ± 1.54	4.17 ± 0.2	7.0 ± 1.06
[L467F;F508del]/ c.3532_3535dup Patient 2	Biopsy № 1	−1.5	0.5	3
	Biopsy № 2	−2.5	0	4
	Biopsy № 3	−2	0.5	3.5
	M ± m	−2 ± 0.35	0.33 ± 0.2	3.5 ± 0.35
[L467F;F508del]/ c.1766+2T>C Patient 3	Biopsy № 1	−3.5	0	3
	Biopsy № 2	0	0	3.5
	Biopsy № 3	−1.5	0	4
	M ± m	−1.67 ± 3.78	0	3.5 ± 0.35
[L467F;F508del]/ W1310X Patient 4	Biopsy № 1	−10	0	9
	Biopsy № 2	−10.5	0	8
	Biopsy № 3	−9	0	9.5
	M ± m	−9.83 ± 0.54	0	8.83 ± 0.54
[L467F;F508del]/ 712-1G>T Patient 5	Biopsy № 1	−11.5	3.5	5
	Biopsy № 2	−5.5	0	2.5
	Biopsy № 3	−5	1.5	2.5
	M ± m	−7.33 ± 2.56	1.67 ± 1.24	3.33 ± 2.7
[L467F;F508del]/ class I		−5.21 ± 0.98	1.23 ± 0.48	5.23 ± 0.67
[L467F;F508del]/F508del *		−11.39 ± 0.79	2.5 ± 0.61	12.39 ± 1.98
F508del/F508del **		−15.7 ± 3.51	3.33 ± 0.63	17.83 ± 3.57
Control group (*wtCFTR*/*wtCFTR*) **		−7.67 ± 1.76	26.72 ± 2.66	109.76 ± 8.18

* [10], ** [11].

## Data Availability

The datasets used and/or analysed during the current study are available from the corresponding author upon reasonable request.

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
