# Peer review of "Estimation of Chloride Channel Residual Function and Assessment of Targeted Drugs Efficiency in the Presence of a Complex Allele [L467F;F508del] in the CFTR Gene"

_ijms, 2024, doi:10.3390/ijms251910424_

Round 1

Reviewer 1 Report

Comments and Suggestions for Authors

This is an interesting case report study about altered CFTR modulator response in patients with cystic fibrosis carrying the complex allele L467,F508del. However, this manuscript needs English language editing to improve the readability.

Critique points:

Introduction: page 1 – line 24-25: please explain what you mean with “independently variants….”.

-              Is there the possibility to provide the actual fecal elastase results, and not only <200

-              For 2 patients, Nanoduct which measures sweat conductivity was performed at baseline on day 1 of treatment and then the Gibson&Cook sweat test was performed at follow-up challenging comparison. Why wasn’t Gibson&Cook performed at baseline?

-              Better to say changes rather than dynamics.

-              Better to provide a) an original trace of the intestinal measurements and b) a summary graph rather than a table to visualize non-response, similarly to the presentation of the organoid results.

-              No need to write all the AUC results in the text.

-              Discussions can be shortened significantly as there is a lot of redundances.

Comments on the Quality of English Language

Tha manuscript needs moderate English editing to improve readability.

Author Response

Response to Reviewer 1 Comments and Suggestions

  1. This is an interesting case report study about altered CFTR modulator response in patients with cystic fibrosis carrying the complex allele L467,F508del. However, this manuscript needs English language editing to improve the readability.

Response 1: Thank you very much! We have edited the manuscript.

Critique points:

  1. Introduction: page 1 – line 24-25: please explain what you mean with “independently variants….”.

Response 2: Corrected (line 34-38).

  1. Is there the possibility to provide the actual fecal elastase results, and not only <200

Response 3. All patients without exception had levels less than 15 ug/g of feces (corrected in Table 1).

  1. For 2 patients, Nanoduct which measures sweat conductivity was performed at baseline on day 1 of treatment and then the Gibson&Cook sweat test was performed at follow-up challenging comparison. Why wasn’t Gibson&Cook performed at baseline?

Response 4.  In regional medical centers the Gibson&Cook was used, as currently in the country there are problems with the using Nanoduct device due to a shortage of consumables.

  1. Better to say changes rather than dynamics.

Response 5. Thank you! Corrected.

  1. Better to provide a) an original trace of the intestinal measurements and b) a summary graph rather than a table to visualize non-response, similarly to the presentation of the organoid results.

Response 6. Thank you for your suggestion. We have added graphs.

  1. No need to write all the AUC results in the text.

Response 7. Thank you very much for your suggestion. We have corrected the text and partially removed the AUC results.

  1. Discussions can be shortened significantly as there is a lot of redundances.

Response 8. Thank you very much, we have shortened Discussions.

  1. Comments on the Quality of English Language

The manuscript needs moderate English editing to improve readability.

Response 9: Thank you very much! We have edited the manuscript.

Reviewer 2 Report

Comments and Suggestions for Authors

Complex alleles have been recognized to affect the molecular function of the respective gene product. In the context of Cystic Fibrosis (CF), complex alleles may contribute to the heterogeneity of the disease phenotype. The manuscript by Anna Efremova describes results obtained from five CF patients that carry the complex allele L467F;F508del in one or both CFTR gene copies. The authors present the patient characteristics and have performed two functional assays (intestinal current measurement, ICM and intestinal organoid swelling) on the patient’s tissue samples.

Irrespective of the individual patient’s mutations, neither ICM nor the intestinal organoid swelling assay showed significant CFTR function. Moreover, treatment of the samples with CFTR modulators could only marginally increase the minimal residual function observed in the swelling assay. The authors reason that the complex allele L467F;F508del aggravates the disease outcome in CF patients while making a treatment with the approved CFTR modulators fruitless. This is also reflected by the lack of clinical response in those patients that had temporarily been put on a therapy with CFTR modulators.

I’m pleased with the contents of this manuscript that deserves publication in IJMS and have only minor points to make:

Line 19: in compound with class I variants -> in combination with class I variants

Line 80: change “Nanodact” to “Nanoduct” (unless the brand name is different in Russia)

Line 99: change “Macrodact” to “Macroduct” (unless the brand name is spelled differently in Russia)

Line 171: the authors correctly state that histamine is believed to activate CaCC channels. In such a case one would expect a positive ISC (due to a flow of Cl- from the inside to the outside of the cell). An inflow of Cations by definition does have the same current direction. Therefore influx of potassium (which is very unlikely to occur in this setting) would also result in a positive current. What is known though, is activation of Ca dependent K+ channels that facilitate potassium secretion (= K+ exit) and results in a negative ISC.

Please review this section.

Comments on the Quality of English Language

Line 19: in compound with class I variants -> in combination with class I variants

Author Response

Response to Reviewer 2 Comments and Suggestions

Complex alleles have been recognized to affect the molecular function of the respective gene product. In the context of Cystic Fibrosis (CF), complex alleles may contribute to the heterogeneity of the disease phenotype. The manuscript by Anna Efremova describes results obtained from five CF patients that carry the complex allele L467F;F508del in one or both CFTR gene copies. The authors present the patient characteristics and have performed two functional assays (intestinal current measurement, ICM and intestinal organoid swelling) on the patient’s tissue samples.

Irrespective of the individual patient’s mutations, neither ICM nor the intestinal organoid swelling assay showed significant CFTR function. Moreover, treatment of the samples with CFTR modulators could only marginally increase the minimal residual function observed in the swelling assay. The authors reason that the complex allele L467F;F508del aggravates the disease outcome in CF patients while making a treatment with the approved CFTR modulators fruitless. This is also reflected by the lack of clinical response in those patients that had temporarily been put on a therapy with CFTR modulators.

I’m pleased with the contents of this manuscript that deserves publication in IJMS and have only minor points to make:

  1. Line 19: in compound with class I variants -> in combination with class I variants

Response 1: Thank you very much for your evaluation of the manuscript! We have made corrections to the text.

  1. Line 80: change “Nanodact” to “Nanoduct” (unless the brand name is different in Russia)

Response 2: Corrected.

  1. Line 99: change “Macrodact” to “Macroduct” (unless the brand name is spelled differently in Russia)

Response 3: Corrected.

  1. Line 171: the authors correctly state that histamine is believed to activate CaCC channels. In such a case one would expect a positiveISC (due to a flow of Cl- from the inside to the outside of the cell). An inflow of Cations by definition does have the same current direction. Therefore influx of potassium (which is very unlikely to occur in this setting) would also result in a positive What is known though, is activation of Ca dependent K+ channels that facilitate potassium secretion (= K+ exit) and results in a negative ISC.

Please review this section.

Response 4: Thank you very much for your careful reading of our article! We made a typo. When we described the results, we relied on the article by Veeze HJ, Sinaasappel M, Bijman J, Bouquet J, de Jonge HR. Ion transport abnormalities in rectal suction biopsies from children with cystic fibrosis. Gastroenterology. 1991 Aug;101(2):398-403. doi: 10.1016/0016-5085(91)90017-f. The article has been corrected (lines 166 and 184).
